# Energy Prediction Models and Distributed Analysis of the Grinding Process of Sustainable Manufacturing

**DOI:** 10.3390/mi14081603

**Published:** 2023-08-14

**Authors:** Yebing Tian, Jinling Wang, Xintao Hu, Xiaomei Song, Jinguo Han, Jinhui Wang

**Affiliations:** 1School of Mechanical Engineering, Shandong University of Technology, Zibo 255049, China; 2Department of Planning and Finance, Shandong University of Technology, Zibo 255049, China

**Keywords:** precision grinding, material removal energy, grinding energy evaluation, energy distribution analysis

## Abstract

Grinding is a critical surface-finishing process in the manufacturing industry. One of the challenging problems is that the specific grinding energy is greater than in ordinary procedures, while energy efficiency is lower. However, an integrated energy model and analysis of energy distribution during grinding is still lacking. To bridge this gap, the grinding time history is first built to describe the cyclic movement during air-cuttings, feedings, and cuttings. Steady and transient power features during high-speed rotations along the spindle and repeated intermittent feeding movements along the x-, y-, and z-axes are also analysed. Energy prediction models, which include specific movement stages such as cutting-in, stable cutting, and cutting-out along the spindle, as well as infeed and turning along the three infeed axes, are then established. To investigate model parameters, 10 experimental groups were analysed using the Gauss-Newton gradient method. Four testing trials demonstrate that the accuracy of the suggested model is acceptable, with errors of 5%. Energy efficiency and energy distributions for various components and motion stages are also analysed. Low-power chip design, lightweight worktable utilization, and minimal lubricant quantities are advised. Furthermore, it is an excellent choice for optimizing grinding parameters in current equipment.

## 1. Introduction

As the engine of economic growth, the manufacturing industry has greatly promoted economic development [1]. However, a series of worldwide environmental problems, e.g., energy shortage and carbon emissions, have resulted [1,2]. According to the International Energy Agency’s report, manufacturing activities account for 17% of global energy consumption and 37% of carbon emissions [3]. There is therefore widespread need to reduce energy and carbon emissions in manufacturing [4]. As the heart of the manufacturing industry, machine tools are the largest contributor to environmental pollution [5] and have great energy-saving potential [6]. Numerous studies have addressed energy characteristics and improvement strategies in various machining processes [7]. Lower energy efficiency—less than 30%—has been identified as the primary cause of the huge waste of energy during machining [8]. Therefore, energy efficiency improvement based on energy modelling and distributed analysis not only merits theoretical research but also has strong practical significance [9].

Liu et al. [10] used spindle speed and input power to measure real-time energy efficiency flow in the spindle system of machine tools without using force measurement equipment. Lv et al. [11] built the acceleration energy model using the principle of spindle mechanical transmission and motor control. Energy was evaluated on the key component level but not on the system level. Energy characteristics and their modelling become complicated with more motion stages and subsystems. Zhou et al. [6] divided the existing system-based energy models of machine tools into three categories: material removal rate, detailed parameter type, and machining process. Zhao et al. [12] proposed predictive energy methods of the process, machine, and system levels based on the power demand of sub-components in different machining states. Lv et al. [13] developed power models of four machining motions: basic, auxiliary, air-cutting, and material removal; however, their versatility is limited.

Shang et al. [14] proposed a generic power model for heavy-duty machine tools for system, framework, and detailed sub-components. Liu et al. [15] presented a generalized method for modelling the inherent energy performance. Power and energy models were built based on a steady process, although this process consumed a larger proportion of energy [16]. Avram et al. [17] developed an energy model to estimate the mechanical energy requirements of different parts of both stable and transient states. Liu et al. [18] proposed a dynamic energy model based on bond graph theory. However, these transient models are system-dependent. Duflou et al. [19] introduced energy-efficient decision-making technologies for energy saving from a unit process focus: multi-machine, factory, multi-facility, and supply chain levels. The energy evaluation results and saving strategies were process- and system-dependent. Similarly, Schudeleit et al. [20] tested the energy efficiency of machine tools using four common methods: reference sample method, reference process method, specific energy consumption method, and component reference method. However, it is difficult to design a set of recognized process standards for reference process methods because different processes may lead to completely different evaluation results. More specifically, energy models for certain machining technologies have been studied.

Commonly used machining methods involve turning, milling, drilling, grinding, and polishing [21,22,23]. Lv et al. [24] compared the prediction accuracy of three models of material removal power during turning, including the specific energy, cutting force, and exponential function method. Yip et al. [25] proposed a modified energy model for ultra-precision diamond cutting where the material recovery effect is considered. Shi et al. [26] introduced an improved cutting power and energy model for milling. Shin et al. [27] proposed a component-based energy modelling methodology for real-time control of milling progress. Four-step procedures were used: data extraction, data pre-processing, context synchronization, and regression modelling. Jia et al. [28] proposed a detailed energy model of the drilling process; power characteristics of all sub-components were considered. Xiao et al. [29] developed a multi-component energy model for dry gear.

Compared with turning, milling, and drilling processes, research on energy models and distribution analyses of grinding and polishing processes are still rare. Compared with grinding, polishing is generally employed after basic cutting processes. Although it is an energy-intensive machining process, surface integrity is the most important process goal. Moreover, many polishing methods are based on energy fields, e.g., submerged pulsating air jet polishing process [30] or long machining time-dependent process [31]. Therefore, energy studies of grinding are more practical [32]. However, the complex material removal mechanism and running process make energy assessment and analysis difficult [33,34]. Dogra et al. [35] comprehensively reviewed the energy-saving and environmental protection method of surface grinding. Wang et al. [36] established production costs, productivity, and CO_2_ emission models for grinding. The energy was considered when evaluating CO_2_ emissions. Deng et al. [37] built a model for energy and carbon efficiency of the wheel spindle using the genetic algorithm. The specific grinding energy and material removal energy of the spindle has been well-modelled [38,39]. In our previous studies, both total and active energies of the spindle were analysed and optimized using machine learning and genetic algorithms [40,41,42]. However, repeated and intermittent infeed movements and high-speed approaching stages consume a large portion of energy and their influence on grinding energy cannot be ignored. Additionally, power laws and time histories for x-, y-, and z-infeed are complex and different. Therefore, an integrated energy model for grinding was built, including all infeed motions and wheel rotation up, down, approaching, and material removal states. Distributed analysis and discussions of grinding energy characteristics were obtained to help managers formulate reasonable optimization schemes for energy-efficient manufacturing.

## 2. Integrated Energy Prediction Model

### 2.1. Energy Flow Analysis during Grinding

Material removal during grinding needs multi-joint motions, except electric controlling and cooling, e.g., wheel rotation along the spindle, and x-, y-, and z-infeed along the x-, y-, and z-axes, respectively. The structure and multi-joint motions of the grinder and energy flow are shown in Figure 1.

Electric controlling components, e.g., lights, chips, and fans start when the power comes on. The cooling system remains automatically open until machining starts. The energy for cooling is consumed by pumps, cooling motors, and air compressors. After standby, the cooling time history runs through the whole machining process. Meanwhile, the rotation and infeed movement start along the spindle and x-axis, respectively. The energy consumption mechanism of the wheel rotation is different in both idle (also called approaching and air cutting) and material removal periods. The x-infeed is driven to move in left and right reciprocating motions. Infeed movement along the z-axis occurs at the end of air cutting in the left and right directions. After a grinding run is complete, the y-infeed starts to drive the wheel to move down a depth of cut. The total energy model can be expressed using Equation (1).
(1)$$E_{\mathrm{total}}=E_{e}+E_{c}+E_{s}+E_{x}+E_{y}+E_{z}$$
where *E*_e_, *E*_c_, *E*_s_, *E*_x_, *E*_y_, and *E*_z_ represent the energy consumed on the electric controller, cooling system, wheel rotation, x-infeed, y-infeed, and z-infeed, respectively.

### 2.2. Energy Modelling of Wheel Air Cutting Rotation and Material Removal

The flow of power of the wheel over time is depicted in Figure 2. Power waves varied periodically. The wave shapes and sizes are determined by the relative locations between the wheel and the workpiece. Two parts are divided into idle motion (also called approaching and air cutting) power and material removal power. The former is associated with rolling friction loss and high-speed rotation of the grinding wheel. Approaching stages are combined in two locations: front and back in position 1 and left and right in position 2. A linear functional relationship with wheel speed is employed to model the approaching power:(2)$$P_{\mathrm{sa}}=A_{\mathrm{sa}}v_{s}+B_{\mathrm{sa}}$$
where *v*_s_ is wheel speed and *A*_sa_ and *B*_sa_ are undetermined coefficients that are associated with wheel motor performance.

The front–back and left–right approaching times during the grinding stroke are computed using Equations (3) and (4), respectively.
(3)$$t_{a1}=\frac{L}{v_{w}}\cdot \frac{4a}{w}$$
(4)$$t_{a2}=\frac{2b}{v_{w}}$$
where *a* and *b* are the front–back and left–right grinding gaps, respectively, *L* is workpiece length, *w* is grinding width, and *v*_w_ is workpiece infeed speed.

Energy in two approaching stages is calculated using Equations (5) and (6).
(5)$$E_{a1}=P_{\mathrm{sa}}t_{a1}m_{z}$$
(6)$$E_{a2}=P_{\mathrm{sa}}t_{a2}m_{z}n_{z}$$
where *m*_z_ is infeed times along the *z*-axis during a grinding stroke and *n*_z_ is the number of grinding strokes.
(7)$$m_{z}=\frac{d}{a_{p}}$$
(8)$$n_{z}=\frac{\left(W+2a\right)}{w}$$
where *d* is the total material removal depth, *a*_p_ is the depth of cut, and *W* is the workpiece width.

The material removal power is divided into three stages: cutting-in in position 3, stable cutting in position 4, and cutting-out in position 5. In position 3, the grinding wheel gradually cuts into the workpiece, which increases the material removal rate. The gradual cutting-in leads to a linear increase in spindle power, like the 3rd identifier in Figure 2a. A linear function with time in Equation (9) is employed to depict the cutting-in power.
(9)$$P_{\mathrm{ci}}=A_{\mathrm{ci}}t+B_{\mathrm{ci}}$$*A*_ci_ and *B*_ci_ are the linear shape coefficient and slope and intercept, respectively.
(10)$$A_{\mathrm{ci}}=\frac{P_{\mathrm{sm}}}{t_{\mathrm{ci}}}$$*t*_ci_ is the time history in the cutting-in stage.
(11)$$t_{\mathrm{ci}}=\frac{\sqrt{R^{2}-{\left(R-a_{p}\times 10^{-3}\right)}^{2}}}{v_{w}}$$*R* is the radius of a grinding wheel.

The cutting-in energy is obtained using Equation (12).
(12)$$E_{\mathrm{ci}}=(P_{\mathrm{sa}}t_{\mathrm{ci}}+0.5P_{\mathrm{ci}}t_{\mathrm{ci}})m_{z}\frac{W}{w}$$

The steady material removal power in position 4 is calculated using Equation (13). It is deduced using the empirical grinding force calculation
(13)$$P_{\mathrm{sm}}=\lambda v_{s}^{\alpha +1}v_{w}^{\beta }a_{p}^{\chi }$$
where *λ*, *α*, *β*, and *χ* are equation coefficients associated with grinding wheel conditions and workpiece properties.

Time history and energy during stable cutting are expressed using Equations (14) and (15), respectively.
(14)$$t_{\mathrm{sb}}=\frac{\left[L-2\sqrt{R^{2}-{\left(R-a_{p}\times 10^{-3}\right)}^{2}}\right]}{v_{w}}$$
(15)$$E_{\mathrm{cu}}=(P_{\mathrm{sa}}+P_{\mathrm{sm}})t_{\mathrm{sb}}m_{z}\frac{W}{w}$$

The material removal rate gradually decreases in the cutting-out stage like position 6. In this stage, power is expressed by a linear function that also takes time into account.
(16)$$P_{\mathrm{co}}=A_{\mathrm{co}}t_{}+B_{\mathrm{co}}$$*A*_co_ and *B*_co_ are the linear shape coefficient and slope and intercept, respectively.
(17)$$A_{\mathrm{co}}=\frac{P_{\mathrm{sm}}}{t_{\mathrm{co}}}$$*t*_co_ is time history during the cutting-out stage.
(18)$$t_{\mathrm{co}}=\frac{\sqrt{R^{2}-{\left(R-a_{p}\times 10^{-3}\right)}^{2}}}{v_{w}}$$

The cutting-out energy is calculated using Equation (19).
(19)$$E_{\mathrm{co}}=(P_{\mathrm{sa}}t_{\mathrm{ci}}+0.5P_{\mathrm{ci}}t_{\mathrm{ci}})m_{z}\frac{W}{w}$$

The energy consumed during the material removal stage is the sum of cutting-in, cutting, and cutting-out energy. The energy model of the spindle is built as the sum of approaching and material removal energy.
(20)$$E_{\mathrm{sm}}=E_{\mathrm{ci}}+E_{\mathrm{cu}}+E_{\mathrm{co}}$$
(21)$$E_{s}=E_{a1}+E_{a2}+E_{\mathrm{sm}}$$

### 2.3. Energy Modelling of x-Infeed

The power waves and relative positions along the x-axis are shown in Figure 3. The x-infeed motion is divided into three stages: acceleration like position 1, constant infeed like position 2, and deceleration like position 3. The maximum power during x-acceleration is associated with the starting characteristics of the x-motor. The cubic function model in Equation (22) is established with workpiece infeed speed as the independent variable.
(22)$$P_{x}=\eta +\xi v_{w}+{\psi v}_{w}^{2}+\omega v_{w}^{3}$$
where *η*, *ξ*, *ψ*, and *ω* are function coefficients.

The constant infeed power along the x-axis is a quadratic function of workpiece infeed speed, as shown in Equation (23).
(23)$$P_{\mathrm{xm}}=A_{x}+B_{x}v_{w}+C_{x}v_{w}^{2}$$
where *A*_x_, *B*_x_, and *C*_x_ are coefficients determined by properties of motor drivers and workbench friction.

The duration of the constant infeed stage is obtained using Equation (24).
(24)$$t_{\mathrm{cx}}=\frac{60\left(L+2b\right)}{v_{w}}-t_{\mathrm{gx}}$$
where *t*_gx_ is the gravity acceleration and deceleration time.

The energy consumed by movement along the x-axis is calculated as follows:(25)$$E_{x}=(P_{\mathrm{xm}}t_{\mathrm{cx}}+0.5P_{x}t_{\mathrm{gx}})m_{z}n_{z}$$

### 2.4. Energy Modelling of y- and z-Infeed

The y-infeed motion is a kind of intermittent feed motion of the grinding wheel along the up and down directions. The y-infeed power, *P*_y_, is regarded as a constant for a certain grinder. The energy model of the y-infeed is expressed using Equation (26).
(26)$$E_{y}=P_{y}t_{y}m_{z}$$
where *P*_y_ is y-infeed power and *t*_y_ is y-infeed time. *m*_z_ is calculated using Equation (7).

Similarly, the energy model of the z-infeed is established using Equation (27).
(27)$$E_{z}=P_{z}t_{z}n_{z}m_{z}$$
where *P*_z_ and *t*_z_ are z-infeed power and time, respectively. *n*_z_ is calculated using Equation (8).

### 2.5. Energy Modelling of the Electric Controller and Cooling System

Electric controlling and cooling powers are modelled as constants in the grinding process; both extend along the whole machining process. The energy models of the electrical and coolant systems are described by Equations (28) and (30), respectively.
(28)$$E_{e}=P_{e}t_{\mathrm{total}}$$
where *P*_e_ is electric controlling power and *t*_total_ is the total time from start-up to shut down.
(29)$$t_{\mathrm{total}}=t_{e}+(t_{z}n_{z}+t_{y}+t_{a2}n_{z}+(t_{\mathrm{ci}}+t_{s}+t_{\mathrm{co}})\frac{W}{w}+t_{a1})m_{z}$$*t*_e_ is the waiting time for the electric control system.
(30)$$E_{c}=P_{c}t_{c}$$*P*_c_ is the cooling power and *t*_c_ is the cooling process time.
(31)$$t_{c}=(t_{z}n_{z}+t_{y}+t_{a2}n_{z}+(t_{\mathrm{ci}}+t_{s}+t_{\mathrm{co}})\frac{W}{w}+t_{a1})m_{z}$$

## 3. Parameters for Grinding Energy Models

### 3.1. Grinding Experiment Setup

Grinding experiments are performed on a three-axis precision grinder (SMART-B818III). The volume of the ceramic composite workpiece (SiO_2f_/SiO_2_) is 50 mm × 50 mm × 25 mm. A diamond grinding wheel with a radius of 100 mm and a width of 10 mm is employed. For the installed wheel, spindle rotating speeds were 10.46 m/s–73.26 m/s (spindle rotating speeds for the grinder were 1000 r/min–7000 r/min). The workpiece infeed speed is best kept within 25 m/min. A water-based hybrid liquid is used for cooling purposes. Grinding parameter settings for this kind of material and wheel, workpiece infeed speed, depth of cut, and wheel speed were investigated in our previous study [43]. The grinding parameters for the three-factor and four-level experimental design are set out in Table 1.

The power for each moving component is measured by the portable power monitoring system. It consists of a power meter (PPC-3), a data acquisition system (NI 9174 and NI 9203), and an analytical tool built in LabVIEW; the hardware structure is shown in Figure 4. PPC-3 has three voltage and current sensors that are connected to the three-phase outputs of the measured system. The response time for PPC-3 is 0.15 s. NI 9203 is used to transfer the analogue power to the digital current signal, which is reverted to the analytical tool.

### 3.2. Experimental Results

The PPC-3 is first connected to the three-phase outputs of the air circuit breaker of the grinder power supply. The stable power is measured after CNC controller start-up. The cooling system and the y- and z-infeed are then opened to record the cooling power, y-infeed power, and z-infeed power, respectively. These steps are repeated five times under different operating conditions. Average power is recorded as *P*_e_, *P*_c_, *P*_y_, and *P*_z_, which are summarized in Table 2. The infeed time along the y-axis, *t*_y_, and the z-axis, *t*_z_, are 0.6 s and 0.3 s, respectively. The gravity acceleration and deceleration time along the x-axis, *t*_gx_, is 0.05 s.

Figure 5a–c shows power curves of wheel rotations corresponding to three stages: start-up, approaching, and material removal. Because the wheel starts once at the short beginning of continuous grinding, the power and energy consumed during the start-up stage are ignored. The variation in power is verified in the former analysis, which has been fully considered.

The power distribution curve of the x-infeed is depicted in Figure 5d–f. Infeed power is the same during the approaching and material removal stages because of the heavy worktable. Power variation in the x-infeed is considered in the modelling section. The energy model was proven to be credible. A total of 10 groups of grinding experiments are designed at 3 factors and 4 levels of *v*_s_, *v*_w_, and *a*_p_. Table 3 shows the grinding parameters and experimental values of wheel rotation power during approaching and material removal stages, as well as x-infeed and x-acceleration power along the x-axis. 

### 3.3. Parameter Studies and Model Verification

Based on the energy models in Section 2, the approaching power of wheel rotation, *P*_sa_, is associated with the unique grinding parameter, *v*_s_. The experimental results in No.1–No.4 are used to obtain the model coefficients, *A*_sa_ and *B*_sa_. A Gauss-Newton gradient method is employed to calculate *A*_sa_ and *B*_sa_ using the reverse gradient. A total of 200 iterations are set. Similarly, the x-infeed power and x-acceleration power are associated with workpiece infeed speed. *v*_w_. *η*, *ξ*, *Ψ*, *ω* in Equation (22) and *A*_x_, *B*_x_, *C*_x_ in Equation (23) are obtained from the fourth to the eighth experimental groups. All 10 experimental groups are required to solve four model parameters for the stable cutting power, *P*_sm_. The coefficients *P*_sa_, *P*_sm_, *P*_xm_, and *P*_x_ in power models are summarized in Table 4.

Another four testing sets are used to verify the accuracy of the energy models. Grinding parameters are stochastically selected for their design requirements. Measured and predicted results are compared in Table 5 and Figure 6. The total forecasting errors of the power models for wheel rotation and x-infeed are estimated within 5%; only one relative error, in the 13th group, reached 4.04%. Excellent prediction accuracy shows that the established grinding energy models are acceptable.

## 4. Energy Distribution Analysis and Discussion

### 4.1. Energy Distribution Analysis and Discussion of Different Components

Energy values of each component from the No.11 to the No.14 verification groups are summarized in Table 6. Energy distributions of the electric controlling system, cooling system, wheel rotation, x-infeed, y-infeed, and z-infeed are shown in Figure 7. The results show that the electrical controller consumes most of the energy during grinding—over 83%; the next consumer is the cooling process (about 11–12%). Moreover, they vary less with grinding parameters. Therefore, the next generation of grinding tool designs should focus on energy-saving chips and cooling pumps. An improved coolant technique, such as dry grinding or minimal quantities of lubricants, is a good choice. Grinding parameters have little influence on total energy distribution but they work on the energy distribution of the motion system, as shown in Figure 7b.

Figure 7b shows larger energy proportions along the spindle wheel and x-axis. Grinding parameters have larger effects on energy distribution; workpiece infeed speed was the most influential variable. A higher speed is desirable for both energy saving and process efficiency improvements. The y-infeed and z-infeed—and particularly the y-infeed—have low energy distributions, i.e., low energy (2.256 J, 0.003% of the total energy) is consumed.

### 4.2. Energy Distribution Analysis and Discussion of Wheel Spindle in Different Machining Stages

Figure 8 depicts the energy consumed by the spindle wheel at different machining stages. Detailed energy consumption along the spindle is summarized in Table 7. The energy and its ratios in front–back approaching, left–right approaching, cutting-in, cutting-out, and stable cutting are analysed.

Figure 8 shows that the stable cutting stage consumes the highest proportion of total energy; longer material removal time contributes to this. The last group of grinding parameters is suggested. The energy ratio in the idle 2 stages is relatively large compared with that in the idle 1 stage. The time in left–right approaching increases significantly due to repeated intermittent feeding. Smaller grinding gaps and bigger grinding widths help reduce energy in approaching stages.

### 4.3. Energy Efficiency Analysis and Improvement Strategies

Energy efficiency is an important optimization objective for CNC machining tools. The proportion of material removal energy is defined as the energy efficiency of grinding problems. Energy prediction results for both stages are shown in Table 8 and the energy ratios are shown in Figure 9.

Figure 9 shows that energy efficiency reaches 36–45%, rising to the general level of mechanical processing (30%). Compared with the results of our previous analysis [40], energy efficiency is improved because the number of grinding strokes is four and not two. Additionally, grinding parameters, particularly for the workpiece infeed speed, have an obvious influence on energy efficiency. A detailed analysis of the optimization of grinding parameters to improve energy efficiency is required in future studies.

## 5. Conclusions

The energy flow process was analysed to distinguish complicated energy characteristics during grinding based on the infeed and material removal grinding paths. An integrated energy model for the whole grinding process, especially the spindle and x-axis systems, was established. Detailed approaching, cutting-in, stable, cutting-out, feeding, and turning stages were described. A total of 14 grinding experiments were designed to study model parameters and verify model accuracy. Energy distribution and efficiency analyses were then performed for new grinder designs and parameter settings. The conclusions are summarized as follows:(1)Predicted power and energy errors compared with measured values were kept within 5%. The integrated energy model is regarded as acceptable for further energy distribution and efficiency analysis.(2)More than 90% of electrical energy is wasted on two auxiliary systems: electrical controlling and cooling. Energy-saving chips, lightweight worktable utilization, and minimal lubricant quantity techniques are recommended in the next-generation design of grinders.(3)Grinding parameters have a larger effect on the energy distribution of both the spindle and the x-axis system. A larger workpiece infeed speed is desired to improve both energy-saving and process efficiency.(4)Energy efficiency reaches 36–45% (over a general level in machining) due to more grinding strokes; it may increase grinding time. A novel balance between energy efficiency, process time, and surface quality should be studied in depth.

There are three limitations to this study. First, the plane grinding process, for example, was thoroughly examined to analyse its energy flow and evaluate energy distribution. Second, stable power characteristics were well examined, while transient situations such as spindle motor start-up, cutting in up-grinding and down-grinding processes, and sparkless grinding were ignored. Despite the low energy efficiency, these transient energy components are valuable for integrated performance improvement of grinding tools. Third, only lower spindle speed was analysed in this study. The power curves of the spindle wheel may differ when idle power is set to medium or high speeds. Thus, it will be critical in the future to examine dynamic and detailed grinding energy flow during other grinding methods, particularly free-form surface grinding.

## 6. Patents

There is a patent resulting from the work reported in this manuscript: a method for evaluating energy efficiency of surface grinding (China, 2022102335044.7).

## Figures and Tables

**Figure 1 micromachines-14-01603-f001:**
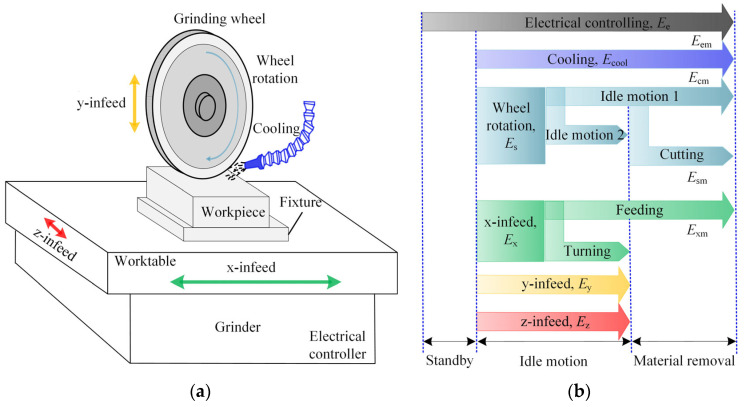
Energy flow during grinding: (**a**) Structural composition of the grinder; (**b**) Energy flow across different machining stages and parts.

**Figure 2 micromachines-14-01603-f002:**
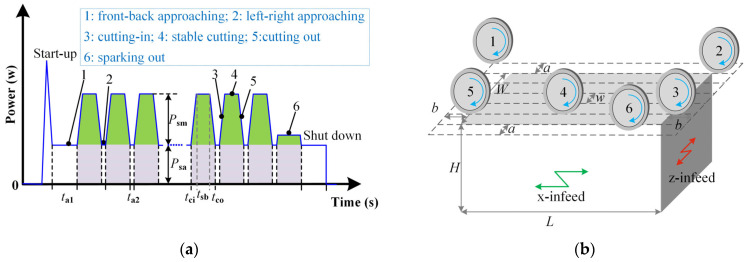
Wheel rotation power during a grinding run: (**a**) Power variation over time; (**b**) Relative position between wheel and workpiece.

**Figure 3 micromachines-14-01603-f003:**
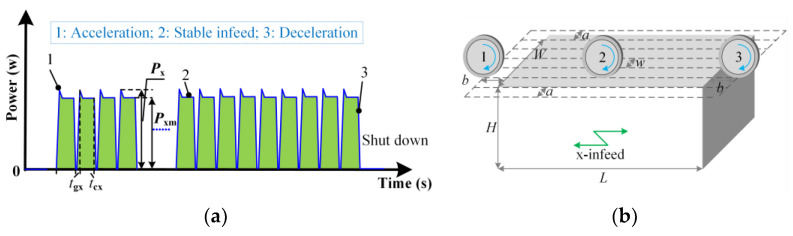
x-infeed power during a grinding run: (**a**) Power variation over time; (**b**) Relative position of the wheel and workpiece.

**Figure 4 micromachines-14-01603-f004:**
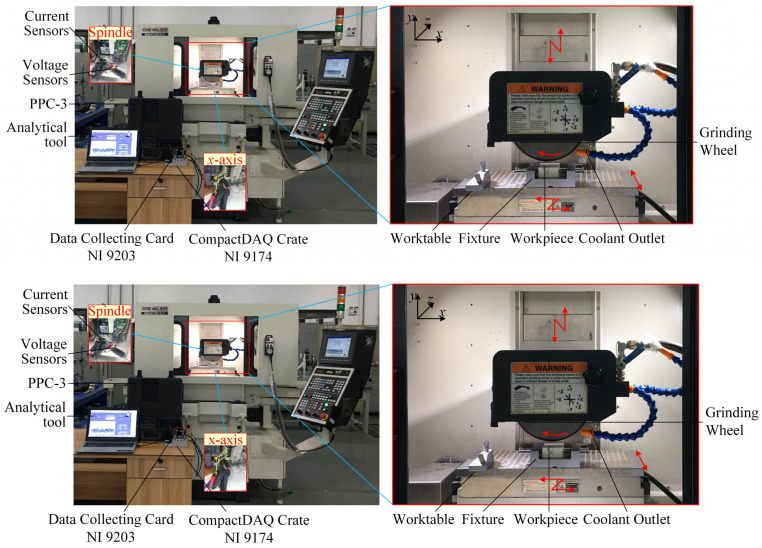
Experimental setup of the grinding power monitoring system.

**Figure 5 micromachines-14-01603-f005:**
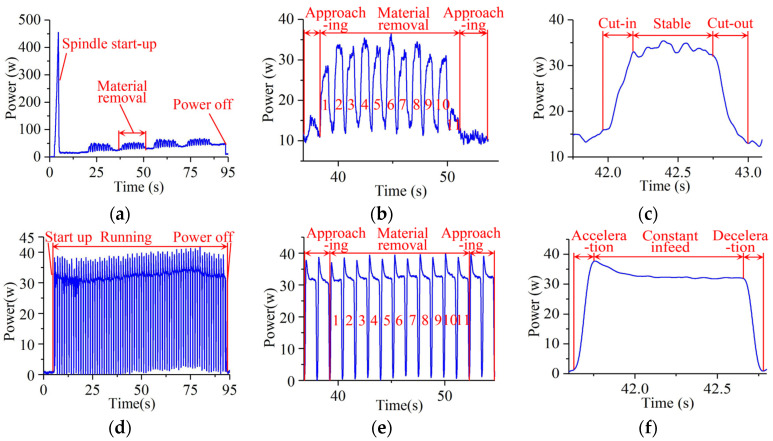
Measured power variation along the spindle and x-axis: (**a**) Power waveform of wheel rotation; (**b**) Rotation power during material removal; (**c**) Rotation power during a grinding stroke; (**d**) Power waveform of x-infeed; (**e**) x-infeed power during material removal; (**f**) x-infeed power during a grinding stroke.

**Figure 6 micromachines-14-01603-f006:**
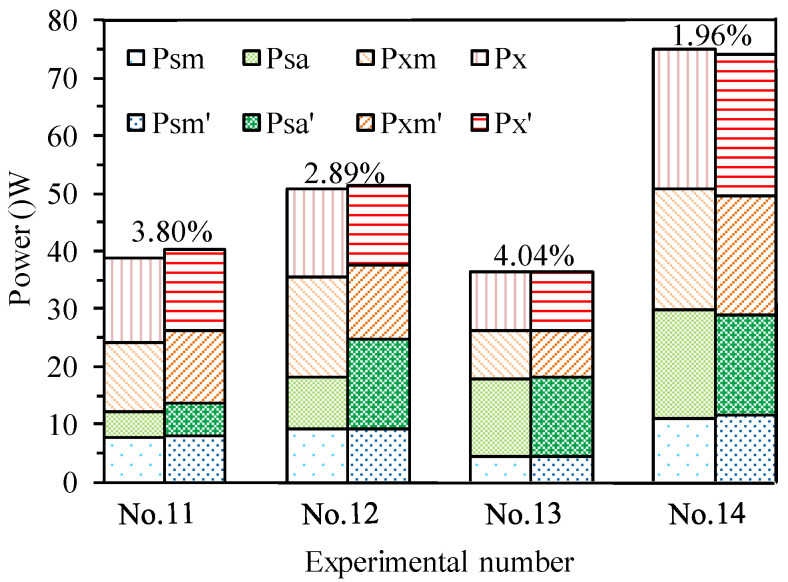
Comparisons between measured and predicted power values of material removal and idle motion along the spindle, acceleration, and infeed along the x-axis.

**Figure 7 micromachines-14-01603-f007:**
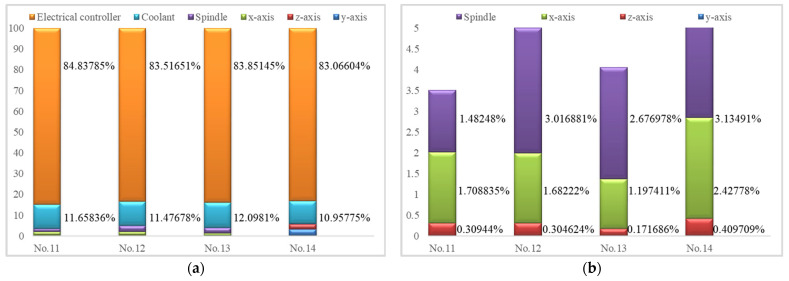
Energy distribution comparisons for different components in four testing groups: (**a**) Energy distribution in the electrical controller, coolant, spindle rotation, x-infeed, y-infeed, and z-infeed parts; (**b**) Energy distribution in four motion parts: spindle rotation, x-infeed, y-infeed, and z-infeed.

**Figure 8 micromachines-14-01603-f008:**
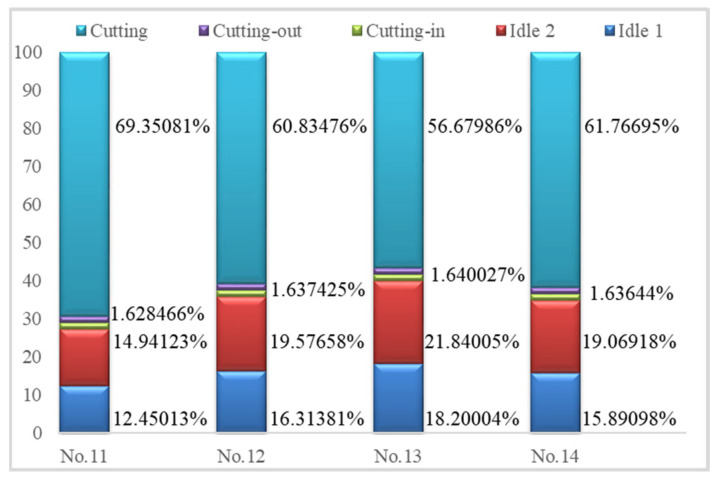
Comparisons of energy distributions in wheel spindle in front–back approaching, left–right approaching, cutting-in, cutting-out, and stable cutting stages.

**Figure 9 micromachines-14-01603-f009:**
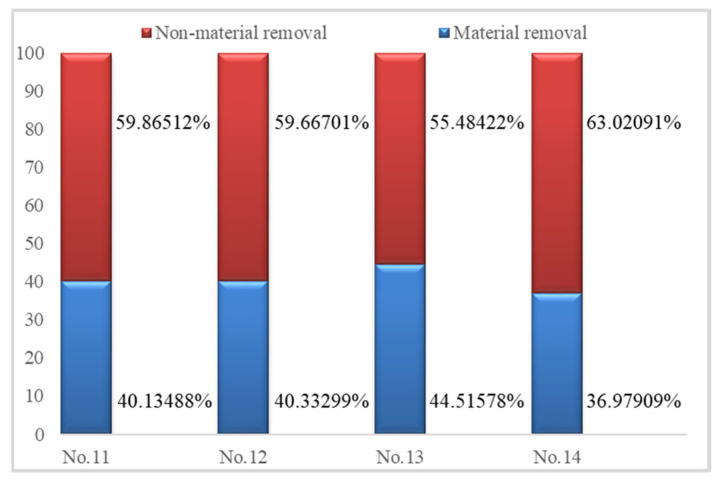
Comparisons of the proportions of energy used in the material removal and non-material removal stages.

**Table 1 micromachines-14-01603-t001:** Grinding conditions.

Factors	Parameters
Machining mode	Plane grinding
Workpiece material	SiO_2f_/SiO_2_ ceramics
Coolant	Water-based
Workpiece size (mm)	50 (L) × 50 (W) × 25 (H)
Grinding wheel geometry (mm)	100 (R) × 10 (Wd)
Grinding width, *w* (mm)	5
Material removal, *d* (μm)	12, 24, 36, 48
Distance, *a* (mm)	5
Distance, *b* (mm)	10
Workpiece infeed speed, *v*_w_ (m/min)	1, 2, 3, 4
Depth of cut, *a*_p_ (μm)	3, 6, 9, 12
Wheel speed, *v*_s_ (m/s)	15, 20, 25, 30

**Table 2 micromachines-14-01603-t002:** Power measurement results of electric controlling, cooling, and y- and z-infeed.

No.	*P*_e_ (W)	*P*_c_ (W)	*P*_y_ (W)	*P*_z_ (W)
1	443.623	65.781	0.923	16.017
2	438.888	67.520	0.947	16.889
3	438.900	66.580	0.947	14.581
4	436.142	67.143	0.997	16.330
5	431.938	68.081	0.889	16.255
Average	437.898	67.021	0.941	16.014

**Table 3 micromachines-14-01603-t003:** Measured results of approaching and material removal power along the spindle and x-infeed and x-acceleration power along the x-axis in 10 experimental groups.

No.	*v*_s_ (m/s)	*v*_w_ (m/min)	*a*_p_ (μm)	*P*_sm_ (W)	*P*_sa_ (W)	*P*_xm_ (W)	*P*_x_ (W)
1	30	4	12	19.77	21.61	31.92	38.63
2	25	4	12	15.58	16.17	31.63	38.28
3	20	4	12	14.97	7.73	31.63	38.53
4	15	4	12	10.65	4.77	31.58	38.52
5	30	1	12	10.16	21.96	8.32	10.27
6	30	2	12	10.70	24.41	11.90	13.81
7	30	3	12	15.09	22.91	21.51	24.43
8	30	4	9	17.90	21.30	31.33	39.86
9	30	4	6	12.60	22.81	31.82	38.30
10	30	4	3	7.72	22.34	31.75	38.94

**Table 4 micromachines-14-01603-t004:** Power model coefficients.

Power	Coefficients			
*P* _sa_	*A* _sa_	*B* _sa_		
	−13.9620	1.1792		
*P* _sm_	*λ*	*α*	*Β*	*χ*
	0.1216	−0.1986	0.6083	0.6014
*P* _xm_	*A* _x_	*B* _x_	*C* _x_	
	6.3500	0.1100	1.5500	
*P* _x_	*η*	*ξ*	*Ψ*	*ω*
	17.22	−13.33	6.947	−0.5679

**Table 5 micromachines-14-01603-t005:** Comparisons between measured and predicted values of material removal and idle motion along the spindle, acceleration, and infeed along the x-axis.

No.	Grinding Parameters			Measured Values				Predicted Values			
	*v*_s_ (m/s)	*v*_w_ (m/min)	*a*_p_ (μm)	*P*_sm_ (W)	*P*_sa_ (W)	*P*_xm_ (W)	*P*_x_ (W)	*P*_sm_^′^ (W)	*P*_sa_^′^ (W)	*P*_xm_^′^ (W)	*P*_x_^′^ (W)
11	16.7	2	12	7.73	4.36	12.19	14.43	7.88	5.73	12.77	13.80
12	25	2	9	9.20	17.25	12.54	13.81	9.16	15.52	12.77	13.80
13	23.4	1	6	4.50	13.43	8.23	10.27	4.47	13.63	8.01	10.27
14	26.7	3	8	11.10	18.67	20.96	24.19	11.52	17.52	20.63	24.41

**Table 6 micromachines-14-01603-t006:** Individual and total energy prediction results for wheel rotation, x-infeed, y-infeed, z-infeed, electrical controller, and coolant.

No.	*E*_s_ (J)	*E*_x_ (J)	*E*_y_ (J)	*E*_z_ (J)	*E*_e_ (J)	*E*_c_ (J)	*E*_total_ (J)
11	1104.487	1273.128	2.256	230.544	63,206.49	8685.792	74,502.69
12	2283.218	1273.128	2.256	230.544	63,206.49	8685.792	75,681.42
13	3594.717	1607.916	2.256	230.544	112,601.6	16,245.65	134,282.7
14	1764.019	1366.116	2.256	230.544	46,741.446	6165.84	56,270.221

**Table 7 micromachines-14-01603-t007:** Individual and total energy prediction results for the wheel spindle in front–back approaching, left–right approaching, cutting-in, cutting-out, stable cutting, and material removal stages.

No.	*E*_a1_ (J)	*E*_a2_ (J)	*E*_ci_ (J)	*E*_co_ (J)	*E*_cu_ (J)	*E*_sm_ (J)	*E*_s_ (J)
11	137.52	165.024	17.9862	17.9862	765.9708	801.9432	1104.4872
12	372.48	446.976	37.386	37.386	1388.9904	1463.7624	2283.2184
13	654.24	785.088	58.95434	58.95434	2037.4808	2155.38948	3594.71748
14	280.32	336.384	28.8672	28.8672	1089.5808	1147.3152	1764.0192

**Table 8 micromachines-14-01603-t008:** Energy predictions for the material removal and non-material removal stages.

No.	*E*_mrr_ (J)	*E*_n-mrr_ (J)	*E*_total_ (J)
11	29,901.564	44,601.1292	74,502.6932
12	30,524.5836	45,156.8408	75,681.4244
13	59,776.98504	74,505.70244	134,282.6875
14	20,808.2168	35,462.0044	56,270.2212

## Data Availability

Not applicable.

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
