# Peer review of "Energy Prediction Models and Distributed Analysis of the Grinding Process of Sustainable Manufacturing"

_micromachines, 2023, doi:10.3390/mi14081603_

Round 1
Reviewer 1 Report
The manuscript presents the energy modeling method and analysis for the surface grinding process. The studies are useful in sustainable manufacturing. Some interesting and new ideas are present in establishing the prediction model and distribution analysis. However, the writing and research results could be improved. I suggest its publication after a revision. The specific comments are as follows:
1. In Introduction, I suggest authors discuss the boundedness of the previous studies, like references [9-12]. Besides, the limitations of your previous studies are suggested to be introduced in detail. In addition, add some important research progress on the polishing techniques, such as studies like Han et al. IJMS, 2023, 257108534 and Han et al. (2020) IJMTM, 156, 103589 .
2. There are some difficult-to-understand descriptions, such as, x-axis feeding and spindle rotation in Figure 1. How the x-axis and spindle is moving?
3. It seems that tm in Figure 2 is not mentioned in next sessions. How it is computed?
4. tcx in Eq.(24) is tcs in Figure 3? I suggest authors check the whole manuscript for this kind of errors.
5. How the parameters in Table 4 are obtained. The method is suggested to be introduced in detail.
6. Try to avoid unnecessary abbreviations, such as "couldn't, won't".
7. I suggest the authors make some discussion for future research and the limitations.
Pls further improve English.
Reviewer 2 Report
In this paper, the energy consumption models of the wheel spindle idling, grinding stage, X,Y,Z axis feed system, electric control and cooling system are given. The energy consumption of the spindle idling is divided into two mathematical models from the left and right position and the front and rear position close to the workpiece, and three mathematical models are established in the grinding stage, namely, the contact workpiece stage, the stable grinding stage and the stage of gradually leaving the workpiece. The whole grinding stage can be regarded as a process of gradual increase in grinding rate from the initial, gradually stable to gradually reduced to zero. The modeling of the X-axis feed system is divided into acceleration, deceleration and uniform speed stages, and the energy consumption model is established by taking the Y and Z axis feed, electric control and cooling system power as constants. Then the experimental verification shows that the error of the model and the experiment is controlled within 5%. Finally, the energy distribution of the grinder is analyzed, and the energy consumption of each part is summarized.
For this article, the following suggestions are given:
1. For spindle modeling, the power change in the transient process of the grinder is rapid and sharp, and the power peak is often caused by the rapid change of mechanical movement and motor start-up in the transient process, and the transient process is frequent in the entire processing process of the grinder, and the total energy consumption caused by it is also not negligible, it is recommended to consider the interaction effect of the transient power.
2. In the actual machining process, there may be different power curves between the spindle rotation speed and the idle power at low, medium and high speed, so it can be considered to set the spindle speed at low, medium and high speed for multiple experiments.
3. In grinding test, the removal rate of added materials can be considered to judge the influence relationship of power.
English language grammer need to be carefully checked in the full paper.
